# A Novel Approach to Improve Acid Diversion in Carbonate Rocks Using Thermochemical Fluids: Experimental and Numerical Study

**DOI:** 10.3390/molecules25132976

**Published:** 2020-06-28

**Authors:** Mustafa Ba Alawi, Amjed Hassan, Murtada Saleh Aljawad, Muhammad Shahzad Kamal, Mohamed Mahmoud, Ayman Al-Nakhli

**Affiliations:** 1College of Petroleum Engineering & Geosciences, King Fahd University of Petroleum & Minerals, Dhahran 31261, Saudi Arabia; g200993130@kfupm.edu.sa (M.B.A.); g201205100@kfupm.edu.sa (A.H.); shahzadmalik@kfupm.edu.sa (M.S.K.); mmahmoud@kfupm.edu.sa (M.M.); 2Advanced Research Center (EXPEC ARC), Dhahran 31311, Saudi Arabia; ayman.nakhli@aramco.com

**Keywords:** acid diversion, thermochemical fluids, in situ gas generation

## Abstract

The distribution of acid over all layers of interest is a critical measure of matrix acidizing efficiency. Chemical and mechanical techniques have been widely adapted for enhancing acid diversion. However, it was demonstrated that these often impact the formation with damage after the acid job is completed. This study introduces, for the first time, a novel solution to improve acid diversion using thermochemical fluids. This method involves generating nitrogen gas at the downhole condition, where the generated gas will contribute in diverting the injected acids into low-permeability formations. In this work, both lab-scale numerical and field-scale analytical models were developed to evaluate the performance of the proposed technique. In addition, experimental measurements were carried out in order to demonstrate the application of thermochemical in improving the acid diversion. The results showed that a thermochemical approach has an effective performance in diverting the injected acids into low-permeability rocks. After treatment, continuous wormholes were generated in the high-permeability rocks as well as in low-permeability rocks. The lab-scale model was able to replicate the wormholing impact observed in the lab. In addition, alternating injection of thermochemical and acid fluids reduced the acid volume 3.6 times compared to the single stage of thermochemical injection. Finally, sensitivity analysis indicates that the formation porosity and permeability have major impacts on the acidizing treatment, while the formations pressures have minor effect on the diversion performance.

## 1. Introduction

Matrix acidizing in carbonates aims to both remove damage near the wellbore area and further improve its permeability by creating conductive flow pathways (wormholes) around the wellbore [1,2,3]. Acid treating fluids designs have diverse compositions and formulations directed towards the targeted formation nature [4,5]. In addition, the acid fluids must be injected below the fracturing pressure to avoid the reservoir to be fractured and prevent the stimulating fluids from being lost consequently [6]. However, the conventional acidizing treatments are less effective with high permeability contrast between zones with large thickness [7,8,9]. With high degree of heterogeneity, it is required that the injected acid treatment is effectively distributed over both the less and more damaged layers [10,11]. Hence, the industry has adapted widely both mechanical and chemical diverters depending on the effectiveness of each approach on the different lithology [12,13].

Mechanical diverters are mainly utilized in sandstone reservoirs and occasionally used for carbonate formations [5,9]. The common mechanical diverters are ball sealers, coiled tubing (CT), and inflow control devices and valves [9,14,15]. Among all the mechanical methods, CT has become a very advantageous tool for enhancing stimulation fluid placement [9]. Acid can be injected through the tubing and terminated conveniently if the treatment is not performing as expected. Additionally, since the size of the tubing is small, the stimulating fluids can be quickly displaced. However, due to the small sizes of CT compared to production tubing and drill pipe, the injection rates can be limited and restricted due the high frictional losses. In addition, corrosion is always with acidizing applications using CT and it is considered disastrous when it is encountered [9]. The mechanical methods are known for effectiveness; however, high costs and risks are always associated whenever they are introduced into the wellbore. Additionally, mechanical diverters such as ball sealers and solid diverting agents are considered as temporary solutions due to their direct effects on downhole temperature that leads them to eventually dissolve.

In addition, there are many designed chemical diverters that are being utilized in the industry [5,7,16]. The injected materials are designed to be present high viscosity contrast to clog the formed wormholes and divert the acid to untreated zones. The widely used chemical diverters are emulsified acids, foams, and in situ gelled acid-based polymers or surfactants [17,18,19,20,21]. Emulsified acids were used to improve the acidizing treatment in high temperature reservoirs by reducing the fluid loss and increasing the acid penetration [22,23]. This has been served by diesel emulsified acid (DEA) which consists of acid, diesel, and an emulsifying agent. The injected emulsion will encroach into the high-permeability layers and divert the acids into the low-permeability layers, utilizing the viscosity contrast in different layers [17].

Moreover, acids can be foamed by introducing gas or by slugging foams into them [16]. Foam enables the placement of a large volume of gas in the formation that is later trapped when acid is injected. Consequently, zones with high gas saturation will clog the pore throats and will require high pressure to be mobilized. Thus, these zones will have low liquid relative permeability that results in acid diverting from them [7]. Recently, nitrogen gas has been adapted for this technique as the discontinuous phase, more so than carbon dioxide, for both economic and environmental reasons. Foam quality is attributed to the volumetric fraction of its gas content, and is an important factor to optimize to attain successful acid diverting performance [21,24]. However, foams require surfactants to keep them stable. In addition, they require the formation to be injected with surfactants as well to ensure its stability in situ [24]. In oil reservoir cases, oil can break the foam stability, so in many cases, it is advised to pump a mutual solvent ahead of the foam [21]. Overall, the main limitation of using foam as an acid diverter is that the foam quality will be significantly reduced when it gets to the wellbore. Therefore, in situ foam generation could be a good solution in order to preserve the advantages of foam treatment and avoid the limitations of foam stability.

Furthermore, viscoelastic surfactant (VES) can be used to divert the injected acid by increasing the fluid viscosity upon the reaction of acid with the formation matrix [20]. The high-viscosity system will block the pathways made inside the rock by the acid and divert it into the uninvaded area to effectively act on the low-permeability zone [25]. Viscoelastic surfactants exhibit efficient and clean wormhole creation inside the treated formation and do not require any flowback processes; solvents or post-flush can be used to retract the viscosity to the gelled acid. However, the effect of this system in causing damage inside the created pathways is possible. In addition, the solvents and VES viscosity contrast can lead to fingering and cause poor formation clean-up [19].

Overall, mechanical approaches have shown good performance in acid diversion, however, they may require high operational cost. In addition, deep or high temperature reservoirs may restrict the application of mechanical diverters due to corrosion issues. On the other hand, chemical diverters provide effective acid placement solutions corresponding to the formation conditions. However, they still have either high potentiality of causing severe damage to the formation or issues in stability; both require further pre- and post-processing. Injection of gas prior to the acid can help significantly in achieving effective acid diversion. However, the major concern is that gas injection may be associated with high operational costs due to gas transportation and treatment. Therefore, in situ gas generation at the reservoir condition can preserve the advantages of using gas injection for acid diversion applications. In addition, this will minimize the operational cost since the gas will be generated only in the wellbore. Similarly, in situ foam generation can result in better foam quality compared to foam injection from the surface. 

This study presents an effective method to place acid uniformly in the damaged formations using in situ generated gas by thermochemical reactions. This method provides the advantage of not requiring an external source of gas for diversion. In addition, the induced pressure from the thermochemical reaction will force the generated gas to flow into the high-permeability zones, thereby improving the performance of acid diversion treatment. In this work, experimental and modeling studies were carried out in order to evaluate the efficiency of the proposed technique. The experiments work constitutes of parallel coreflooding tests using rock samples that have similar geometry and petrophysical properties but different permeability values to simulate heterogeneous formations. A new analytical model was developed to predict the performance of acid diversion using thermochemical fluids. Finally, sensitivity analysis was conducted to assess the impact of formation parameters on the treatment efficiency. The influences of formation porosity, layer permeability, and pressure difference between the two layers on the acid diversion performance were studied.

## 2. Results and Discussion

### 2.1. Impact of Acid Diverter on Acidizing Performance

#### 2.1.1. Experimental Outcomes

Acidizing experiment was conducted using two rock samples with permeability contrast to study the acid distribution when the acid is injected solely without any diverter. Two carbonate rocks were treated with 15 wt% HCl acid, and the used samples had initial permeability values of 90 and 27 mD, respectively. Parallel coreflood system was used, and the two rock samples were treated using the same experimental conditions, with a similar injection pressure and acid flow rate applied to the two rock samples. However, when acid was injected, the majority of the acid invaded into the high-permeability core (90 mD) and only small amounts flowed into the lower permeability core (27 mD). Therefore, a continuous wormhole was generated in the high-permeability rock, while limited wormhole generation was induced in the low-permeability rock. Figure 1 shows the CT-scan images of the tested rocks after acidizing treatment. It should be noted that this acidizing treatment was conducted without using any acid diverter. Therefore, the propagation of wormholes in the high-permeability core achieved breakthrough successfully while it did not in the low-permeability core. Furthermore, acidizing treatment was performed, and thermochemical fluids were injected to witness its diversion performance. Two rock samples with original permeability of 94 and 26 mD were used in this experiment. Thermochemical fluids were injected into the parallel core system prior to acid treatment. In this case, the injected acid was diverted to the lower permeability core. Therefore, the high- and low-permeability rocks were successfully stimulated. Figure 2 shows CT-scan images for the treated rocks after the treatment using HCl acid and thermochemical fluids. Continuous wormholes were observed in the high- and low-permeability rocks, indicating the effective performance of thermochemical fluids in diverting the injected acids into the low-permeability rocks.

#### 2.1.2. Model Match

The abovementioned experimental conditions were used as input in the model to replicate the impact of permeability contrast on acid treatment. The cross-sectional area of the core was modeled to be square (1.33 in edge length), which has a similar area to the 1.5 in diameter core. The model assumes that the two core samples were placed on top of each other with a thin low-permeability layer in the middle acting as a flow barrier. This is to account for the actual separation between the cores in the experimental setup. The porosity and permeability were randomly populated in the domain using Gaussian distribution. Both cores were subjected to the same 15 wt% HCl acid which was injected at 1 cc/min. Figure 3 shows the permeability distribution where the high-permeability core sample was placed on the top. The mean permeability of the top domain was 90 mD, while it was 27 mD for the bottom domain; this was to replicate the behavior in Figure 1. The porosity distribution of both samples is shown in Figure 4 where the mean porosity is 0.2. Porosity lab measurements of the high- and low-permeability rock samples were similar; hence, the same distribution was assigned during simulations. Some statistical parameters, such as the correlation length (*I_x_*, *I_y_*), were assumed to be similar for both porosity and permeability. The standard deviation determines how far the distribution is from the mean. The standard deviation for permeability distribution in the domain, $\sigma_{k}$, was 3, while for porosity, $\sigma_{\varphi }$ was 0.04, as illustrated in the figure captions. 

Figure 5 shows the final concentration profile after wormhole breakthrough assuming HCl acid was injected with no diversion. It is noted that wormhole propagated fully in the high-permeability core (the top domain of Figure 5) while only limited propagation was observed in the low-permeability core. Various simulations of different permeability contrast were conducted, and all showed that acid preferentially invaded the high-permeability core. This replicated the experimental outcomes obtained in Figure 1 which also built confidence in the fine-scale modeling approach.

Once the thermochemical fluids react, they generate a tremendous amount of nitrogen that invades the high-permeability zone, diverting acid to the low-permeability one. Modeling the diversion impacts of thermochemical fluids requires a two-phase flow model which is out of the current research scope. Nevertheless, the simulations in Figure 6 show that the wormhole propagated equally only when the permeability of the two cores to the HCl acid was within the same range. This indicates that the equal acid stimulation in Figure 2 was generated due to the thermochemical fluids’ ability to equate the relative permeability of the two cores.

### 2.2. Alternating Injection of Thermochemical and Acid Fluids

The thermochemical approach showed good performance in diverting the injected acids and then enhancing the efficiency of acidizing treatment. However, the treatment efficiency can be further improved by alternately injecting thermochemical and acid fluids. The alternating injection can lead to a reduced amount of injected fluids and it can also result in deeper acid penetration compared to the single-stage injection. Therefore, in this work, the technique of alternating injection of thermochemical and HCl fluids into the rock samples of different permeability (11 and 102 mD) was examined. A total of six injection cycles were applied: three thermochemical fluid and three HCl injection cycles. Figure 7 and Figure 8 show the pressure profiles for the low- and high-permeability rocks, respectively. The duration of each cycle was determined based on the stability of the pore pressure along the treated rock. During the thermochemical cycles, the pressures at the cores’ inlet and outlet are almost flat for all samples. However, during the HCl injection, the pressures fluctuate due to the reaction between the injected fluid and the carbonate matrix. At the final stage of acid injection, the pressure fluctuation is very high, which can indicate significant solid migration. Injecting HCl into carbonate rock can dissolve part of the carbonate matrix and induce solid migration [26]. Moreover, Figure 9 compares the pressure-drop profiles for the low- and high-permeability rocks during the alternating injection of thermochemical and HCl fluids. The pressure-drop profiles indicate that wormholes were generated in both rocks, since the final pressure-drop is almost zero. The wormhole was created in the high-permeability rock during the second cycle, while it was generated during the fourth injection cycles in the low-permeability rock. Ultimately, continuous wormholes, from the core inlet to the outlet, were achieved in both samples after the thermochemical/HCl treatment. Figure 10 and Figure 11 show images for the cores’ inlet and outlet after the alternating injection of thermochemical and HCl fluids.

### 2.3. Modeling of Acid Diversion Using Thermochemical Fluids 

#### 2.3.1. Single-Stage Injection

The developed model was used to assess the acidizing treatment in a heterogenous reservoir with two layers of different permeability values (20 and 100 mD). The injected acid will be diverted into the low-permeability layer utilizing the in situ gas generation technique. The description of wellbore and reservoir formations during this process are provided in Section 3.1 and Section 3.2. The properties of the reservoir, well, and fluid are listed in Table 1. The top layer is characterized by considerably high permeability compared to the bottom layer. Table 2 summarizes the properties of top formation (layers 1) and bottom formation (layer 2). The permeability of layer 1 is five times higher than the permeability of layer 2. Therefore, during the thermochemical injection, most of the generated gas will invade into layer 1. In this work, the performance of thermochemical fluids in diverting the injected acids was quantified by determining the flow rates into the top (Q1) and bottom (Q2) layers. Then, the ratio Q1/Q2 was used as an indicator of diversion efficiency. If the Q1/Q2 ratio equals one, this indicates that the injected fluids are equally distributed into the two layers. Meanwhile, if the flow rate ratio is less than one, this reveals that the injected acid was successfully diverted into the low-permeability layer [8], and the acid flow rate into the low-permeability layer (Q2) is higher than into the high-permeability layer (Q1).

Figure 12 shows the profiles of Q1/Q2 ratio as a function of the injected acid volume, with thermochemical reaction and N_2_ injection used as acid diverter. The flow rate ratio (Q1/Q2) is decreasing, with acid injection indicating that the injected acid is invading into the low-permeability layer and this will result in reducing the skin factor in the low-permeability layer. Figure 13 shows the formation skin for the top and bottom layers against the acid volume. This figure indicates that thermochemical reaction has a better diversion performance than nitrogen injection since the thermochemical reaction showed lower skin values compared to the N_2_ injection. Thermochemical approach can reduce the acid volume by around 23% compared to the N_2_ injection, and the required acid volume was reduced from around 22 to less than 18 ft^3^/ft. This can be attributed to the injection of thermochemical fluids providing more pressure difference and viscous contrast compared to the N_2_ injection. Ultimately, the thermochemical approach showed effective acid diversion as indicated by the reduction in the Q1/Q2 ratio to less than one, revealing that more acid is invaded into the low-permeability layer.

#### 2.3.2. Alternating Thermochemical and Acid Injection

Alternating thermochemical and acid fluids injection can be used to improve the efficiency of acid diversion and minimize the amount of injected acids. The developed model was used to predict the performance of alternating thermochemical and acid injection. Figure 14 shows the reduction in skin factors in high- and low-permeability layers during the alternating thermochemical and acid injection. The alternating injection approach showed better performance compared to the single stage of thermochemical injection, with the treatment volume reduced from 20 to around 5.6 ft^3^/ft. Alternating the thermochemical fluids will advance the treatment by the advantages of increasing the pressure and temperature in every cycle and then reduce the acid volume by around 3.6 times compared to single-stage injection.

### 2.4. Sensitivity Analysis

Sensitivity analysis was conducted to assess the impact of formation parameters on the treatment efficiency. The influences of formation porosity, layer permeability, and pressure difference between the two layers on acid diversion performance were studied. Table 3 lists the model parameters of the different cases used in the sensitivity analysis work. Figure 15 shows the impact on the pressure difference on the acid diversion performance. The pressure difference between the two layers (low- and high-permeability layers) varied from 600 to 3100 psi, and the diversion performance was evaluated based on the Q1/Q2 ratio. Recalling that, a lower Q1/Q2 value indicates that the injected acids are invading into the low-permeability layers and better acid diversion is then obtained. The pressure difference showed a minor effect on the diversion performance. Similar profiles of Q1/Q2 were observed using different pressure values in the treated formations. This is caused by the major difference of viscosity between the reservoir liquid and injected gas. Thus, it has substantially overcome the effect of pressure difference between the reservoir and bottomhole ones. Although pressure difference is a major contributor to the injected fluids to encroach further to the wellbore adjacent layers, the model honors the viscosity contrast and adequately illustrates almost equal flowrates to both layers.

Additionally, Figure 16 examines the response of diversion performance with different porosity values. The highest porosity values exhibited the best diversion performance compared to the other two cases. The bigger porosity value with gas invading the top layer pores assisted in providing an additional drive for the acid to be diverted to the lower layer. By contrast, the smaller porosity value allowed for small volumes that acid can saturate and, consequently, the diversion effect was slightly less. Moreover, the effect of layer permeability on the acid diversion performance is shown in Figure 17, where permeability ratios of 5 and 50 were used. As the results convey, the model manifested permeability as the major driving parameter for acid flow between the two layers. As the permeability of the top layer increases, it accumulates most of the injected acid correspondingly. This has also resulted in delaying the treatment of the bottom layer massively in comparison to the case with less permeability value of the top layer. Finally, while the bottom layer permeability was decreasing down to the tight levels, and the diversion effect responded with weaker performance. Additionally, the small permeability draws slight volumes of acid resulting in the top layer drawing most of this. As an ultimate response, the top layer gets a significant lead to be treated ahead of the bottom layer.

## 3. Methods and Materials 

### 3.1. Description of the Proposed Approach

Thermochemical fluids are used to improve acid diversion, utilizing the in situ gas generation mechanism. Figure 18 shows a schematic of the acid diversion using thermochemical fluids, where two layers of different permeability are shown (K_1_ and K_2_). Initially, the two layers are saturated with oil. Then, thermochemical fluids are injected to react downhole and generate nitrogen gas. Due to the flow resistance, most of the generated gas will flow into the high-permeability layer while a small amount will flow into the low-permeability zone. The chemical reaction can be triggered by the reservoir temperature and acid can also be injected to reduce the solution pH and accelerate the thermochemical reaction [27,28]. Thereafter, acid is injected to stimulate the reservoir layers and reduce the formation skin. During the acidizing treatment, the generated nitrogen will block the high-permeability zones and divert the injected acids into the low-permeability zones. Consequently, both the high-permeability and low-permeability zones will be stimulated. 

The thermochemical fluids reaction can be represented by Equation (1) [27,28]:(1)$$NH_{4}Cl+NaNO_{2}\to NaCl+2H_{2}O+N_{2}\;\left(g\right)+\Delta H\left(heat\right)$$

Nitrogen gas (N_2_) will be generated in a form of chemically induced pulse that ought to significantly increase the pressure of the invaded zones, and the generated pressure can reach up to 5000 psi [29]. The induced pressure in the wellbore will force the generated gas to flow into the reservoir formations. Due to the flow resistance, most of the generated gas will flow into the high-permeability zones while small volume of N_2_ gas will invade into the low-permeability zones. Hence, the generated gas will promote the acid to be diverted toward the low-permeability layers for which treatment is desired. Additionally, the thermochemical reaction will increase the temperature substantially, and could reach up to 600 °F [29], which will reduce the viscosity of the formation fluids. This will also promote the viscous contrast diversion to distribute the acid efficiently over the formation zones. However, the thermochemical treatment should be precisely designed in order to optimize the amounts of generated gas and to ultimately attain successful treatment. The volumes and concentrations of thermochemical fluids should be accurately selected to ensure that the generated pressure and temperature will be within the specifications of downhole completions. 

### 3.2. Model Development

#### 3.2.1. Numerical Fine-Scale Model 

A finite volume based average continuum, also known as the two-scale model, was developed in this research to observe the impact of permeability contrast on acid distribution [30]. It is assumed that at any point in the domain, both fluids and solids exist in a proportion defined by the porosity. The elements in the domain are large enough to contain both solids and liquids but small enough to have homogenous properties within the element. A correlated random Gaussian distribution of porosity and permeability was implemented to ensure a heterogonous system within the full domain.

First, the continuity equation is implemented which conserve mass in the liquid phase, expressed as
(2)$$\nabla \cdot \left(\rho_{f}u\right)=-\frac{\partial \left(\phi \rho_{f}\right)}{\partial t}$$
where $\rho_{f}$ is the fluid density, u is the velocity vector, ϕ is the porosity, and t is time. The superficial velocity defined by Darcy’s law is written as
(3)$$u=-\frac{k}{\mu_{f}}\cdot \nabla p$$
where k is the permeability tensor,$\;\;\mu_{f}$ is the fluid viscosity, and p is the pressure. The fluid superficial velocity is obtained from solving the system of equations above. Then, the convective, diffusive, and reactive transport equation of acid in porous media is solved:(4)$$\frac{\partial (\phi \rho_{f}C_{A})}{\partial t}+\nabla .\left(\rho_{f}uC_{A}\right)=\nabla .\left(\phi \rho_{f}D_{e}\nabla C_{A}\right)+a_{v}\frac{\left(k_{r}k_{c}\right)}{k_{r}+k_{c}}C_{A}$$
where $C_{A}$ is the HCl acid concentration represented in mass fraction, $D_{e}$ is the effective acid diffusion, $k_{r}$ is the reaction rate constant, $k_{c}$ is the mass transfer coefficient, and $a_{v}$ is the carbonate mineral-specific area. The first term in the equation above represents acid accumulation, the second term represents acid convection, the third term represents acid diffusion, and the last term represents acid/rock reaction. The acid/rock reaction causes the porosity of the rock to increase according to the following:(5)$$\frac{\partial \left(\phi \right)}{\partial t}=a_{v}\frac{\left(k_{r}k_{c}\right)}{k_{r}+k_{c}}C_{A}\beta_{100}(\rho_{f}/\rho_{r})$$
where $\beta_{100}$ is the dissolving power of the full strength HCl acid, and $\rho_{r}$ is the rock density.

##### Constitutive Equations

The algorithm used in this model is sequential, which starts by solving Equations (2) and (3) for the velocity profile, Equation (4) for the acid concentration, and Equation (5) for acid/rock dissolution. To solve the system above, some properties should be updated at each time step, such as $k_{c}$, $D_{e}$, $a_{v}$, and k. The effective acid diffusion in porous media can be related the molecular diffusion,$\;D_{m}$, through the following:

(6)$$D_{e}=\phi D_{m}$$

The mass transfer coefficient can be obtained through relating Schmidt number, Sc, Reynold number of flow in pores, $Re_{p}$, and Sherwood number, Sh. The following are the definitions of these numbers:(7)$$Sh=Sh_{\infty }+bRe_{p}^{\frac{1}{2}}\;Sc^{\frac{1}{3}}$$
(8)$$Re_{p}=\frac{2\rho_{f}r_{p}\left|u\right|}{\mu_{f}}$$
(9)$$\mathrm{Sc}=\frac{\mu_{f}}{\rho_{f}D_{m}}$$
where $r_{p}$ is the pore radius, $Sh_{\infty }$ is the asymptotic Sherwood number, and b is a constant. The mass transfer coefficient can be obtained by solving the equation below once Sherwood number is determined from Equation (7).
(10)$$Sh=\frac{2k_{c}r_{p}}{D_{m}}$$

The equations below are used to update the pore radius, permeability, and specific surface area based on the change in porosity [31]:(11)$$r_{p}=r_{p}{}_{o}{\left(\frac{\phi }{\phi_{o}}\right)}^{\varepsilon }$$
(12)$$k=k_{o}{\left[{\left(\frac{\phi }{\phi_{o}}\right)}^{3}{\left(\frac{1-\phi_{o}}{1-\phi }\right)}^{2}\right]}^{\gamma }$$
(13)$$a_{v}=a_{v}{}_{o}{\left[\left(\frac{\phi }{\phi_{o}}\right)\left(\frac{1-\phi_{o}}{1-\phi }\right)\right]}^{\eta }$$
where ε, γ, and η are empirical parameters that could be used to tune the model with experiment outcomes. 

##### Initial and Boundary Conditions

To solve Equations (2) and (4), initial and boundary conditions should be implemented. The following conditions were implemented for Equation (2):(14)$$p\left(x,y,z,\;t=0\right)=p_{i}\;Initially$$
$${\left(\frac{\partial p}{\partial x}\right)}_{\left(x=0,y,z,t\right)}=-\frac{qB\mu_{f}}{k_{x}A_{c}}\;at\;the\;inlet\;p\left(x=L,y,z,\;t\right)=p_{out}\;at\;the\;outlet\;n.\nabla p=0\;at\;the\;other\;boundaries$$
where $p_{i}$ is the initial pressure, $p_{out}$ is the outlet pressure, q is the injection rate, B is the formation volume factor,$\;A_{c}$ is the grid cross-sectional area, L is the core length, and n is the normal vector to the surface. The conditions for Equation (4) are as follows: (15)$$C_{A}\left(x,y,z,\;t=0\right)=0\;Initially$$
$$C_{A}\left(x=0,y,z,\;t\right)=C_{ini}\;at\;the\;inlet\;n.\nabla C_{A}=0\;at\;the\;other\;boundaries$$
where $C_{ini}$ is the initial acid concentration. 

#### 3.2.2. Skin Revolution Model

An analytical model was developed to evaluate the performance of using thermochemical fluids to improve the acidizing treatment in heterogenous reservoirs. MATLAB program was developed to estimate the skin factor for the permeable and tight zones around the wellbore. The skin factor for each zone will be determined as a function of the injected acids, assuming incompressible and immiscible fluids. In addition, due to the substantial mobility ratio of the generated gas with comparison to the injected acid, the flow of acid is assumed to be acting with piston-like displacement. Moreover, it is assumed that nitrogen gas is encroaching the permeable layer in significant amounts, while negligible amounts of the nitrogen gas will invade into the low-permeability layer. Finally, the layer skin is used as an indicator of the acidizing performance, with the layer skin being reduced with higher volumes of acid invading into the treated layer. The equations used to develop the acid diversion model are described below. 

The pressure difference due to formation skin can defined as follows [32]: (16)$$\Delta p_{s}=\Delta p_{real}-\Delta p_{ideal}$$
where $\Delta p_{s}$ is the pressure drop due to skin, $\Delta p_{real}$ is the real or actual pressure drop, and $\Delta p_{ideal}$ is the ideal pressure drop where no damage was induced. The pressure drop can be determined assuming steady state flow as follows:(17)$$\Delta P=\frac{q\mu }{2\pi kh}\ln\left(\frac{r_{e}}{r_{w}}\right)\;$$
where q is the flow rate, *μ* is the fluid viscosity, *k* is the formation permeability, *h* is the formation thickness, *r_e_* is the drainage radius, and *r_w_* is the wellbore radius. Figure 19 shows the different regions around the wellbore during acidizing treatment. 

Considering the regions around the wellbore that are saturated with spacer and acids fluids, the pressure drop for real and ideal cases can be expressed as:(18)$$\Delta P_{real}=\frac{q\mu_{s}}{2\pi kh}\ln\left(\frac{r_{s}}{r_{a}}\right)+\frac{q\mu_{a}}{2\pi kh}\ln\left(\frac{r_{a}}{r_{w}}\right)$$
(19)$$\Delta P_{ideal}=\frac{q\mu }{2\pi kh}\ln\left(\frac{r_{s}}{r_{w}}\right)$$
where *r_s_* is the distance to which spacer penetrated, *r_a_* is the acid-saturated zone,$\;\mu_{s}$ is the viscosity of spacer fluid, and $\mu_{a}$ is the acid viscosity. Substituting Equations (16) and (17) in Equation (18), the pressure drop due to formation skin can be given by
(20)$$\Delta P_{s}=\Delta P_{real}-\Delta P_{ideal}=\frac{q\mu_{s}}{2\pi kh}\ln\left(\frac{r_{s}}{r_{a}}\right)+\frac{q\mu_{a}}{2\pi kh}\ln\left(\frac{r_{a}}{r_{w}}\right)-\frac{q\mu }{2\pi kh}\ln\left(\frac{r_{s}}{r_{w}}\right)$$

Since the treated zones have similar thickness and flow rate, the pressure drop can be expressed as
(21)$$\Delta P_{s}=\frac{q\mu }{2\pi kh}\left[\frac{\mu_{s}}{\mu }\ln\left(\frac{r_{s}}{r_{a}}\right)+\frac{\mu_{a}}{\mu }\ln\left(\frac{r_{a}}{r_{w}}\right)-\ln\left(\frac{r_{s}}{r_{w}}\right)\right]$$

Then, the skin factor ($S_{vis}$) can be defined as
(22)$$S_{vis}=\left[\frac{\mu_{s}}{\mu }\ln\left(\frac{r_{s}}{r_{a}}\right)+\frac{\mu_{a}}{\mu }\ln\left(\frac{r_{a}}{r_{w}}\right)-\ln\left(\frac{r_{s}}{r_{w}}\right)\right]\;\;\;\;$$
and the pressure drop due to formation skin will be given as
(23)$$\Delta P_{s}=\frac{q\mu }{2\pi kh}S_{vis}.$$

Moreover, in this work, the diversion effect was quantified as a viscous skin factor *S_vis_* for gas with viscosity *µ_g_* that is being displaced by an acid with viscosity *µ_acid_* as the following [32]:(24)$$S_{vis}=\left(\frac{\mu_{acid}}{\mu_{g}}-1\right)\ln\frac{r_{acid}}{r_{w}}.$$

The radius of acid penetration *r_acid_* towards the formation targeted zone can be calculated as
(25)$$V_{acid}=\pi \phi h\;\left(r_{acid}^{2}-r_{w}^{2}\right)$$
where *V_acid_* is the volume of injected acid, ϕ is the formation porosity, and *h* is the formation thickness. Then, the radius of acid penetration (*r_acid_*) can be determined by rearranging Equation (25).
(26)$$r_{acid}=\sqrt{r_{w}^{2}+\frac{{\bar{V}}_{j}}{\pi \phi }}$$
where ${\bar{V}}_{j}$ is volume per unit thickness (*V*/*h*) of acid invaded into a particular zone (*j*). Equation (26) can be substituted in Equation (24) to get
(27)$$S_{vis}=\left(\frac{\mu_{acid}}{\mu_{g}}-1\right)\left[0.5\ln\left(r_{w}^{2}+\frac{{\bar{V}}_{j}}{\pi \phi }\right)-\ln r_{w}\right]$$

Equation (27) indicates that the viscous skin factor ($S_{vis}$) is changing with the volume of the injected acid (${\bar{V}}_{j}$). Assuming steady state flow, the acid flow rate into a particular layer (*q_i_*) can be determined as follows [8]:(28)$$\frac{q_{i}}{h}=\frac{4.92\times 10^{-6}k\left(P_{w}-P_{e}\right)}{\mu \left(\ln\left(\frac{r_{e}}{r_{w}}\right)+S+S_{vis}\right)}.$$

Then, for a certain zone, the cumulative volume of injected acid can be estimated utilizing the acid flow rate (*q_i_*) as follows: (29)$$V_{1}=V_{0}+{\left(\frac{q_{i}}{h}\right)}_{0}\left(t_{1}-t_{0}\right)$$

Acid-injected volumes, in turn, will be increasing until the removal of all damage, at which point the formation skin (*S*) can be deduced depending on the acid volume (*V_c_*) [8]:(30)$$S=S_{0}-C\;{\bar{V}}_{c}$$
where *S*_0_ is the original skin, *C* is the rate of skin decrease, and ${\bar{V}}_{c}$ is the cumulative injected acid at a particular time.

Ultimately, the developed model can estimate the formation skin in each layer utilizing the radius of the acid and spacer zones around the wellbore as presented in Equation (22). In addition, the developed model can estimate the formation skin as a function of the injected acid and the original formation skin (before treatment) as presented in Equation (30). Therefore, MATLAB program was developed to estimate the skin factor using Equations (22) and (30).

### 3.3. Experimental Measurements 

Coreflooding experiments were conducted in order to determine the performance of the developed approach of using thermochemical fluids in diverting the injected acids. A parallel coreflood setup was used, and carbonate rock samples of different permeability values were used. The acidizing experiments were conducted under high pressure and high temperature conditions. An overburden pressure of 1000 psi and backpressure of 400 psi were applied. In addition, HCl of 15wt% concentration was used, the injection rate was maintained at 1 cm^3^/min, and the system temperature was 80 °C. Moreover, the rock samples were prepared by measuring the rock permeability and porosity. Then, the treatment was implemented using alternating injection of HCl acid and thermochemical fluids. The pressure drop across the high- and low-permeability rock samples were monitored using pressure transducers at the core’s inlet and outlet. The properties of the used rock samples are listed in Table 4, and Figure 20 shows the experimental setup used in this work. 

## 4. Conclusions

This study presents a novel solution to improve acid diversion utilizing in situ nitrogen generation. This method involves injecting thermochemical fluids during the acidizing treatment. In this work, numerical and analytical models were developed, and experimental measurements were carried out in order to evaluate the performance of the proposed technique. Based on this study, the following conclusions were drawn:Injecting thermochemical fluids before acidizing treatment showed an effective performance in diverting the injected acids into the low-permeability rocks and then enhancing the efficiency of acidizing treatment.After the treatment, CT-scan images showed that continuous wormholes were generated in the high-permeability rocks as well as in low-permeability rocks.The two-scale continuum model was able to reproduce the experimental outcomes of wormhole propagation.Alternating injection of thermochemical and acid fluids showed better acidizing performance compared to the single stage of thermochemical injection.Alternating the thermochemical fluids will advance the treatment through the advantages of increasing the pressure and temperature in every cycle and then reduce the acid volume by around 3.6 times compared to single-stage injection.The thermochemical approach showed better diversion performance compared to nitrogen injection, and lower values of formation skin were obtained using the thermochemical fluids.Injecting thermochemical fluids can provide greater pressure difference and viscous contrast compared to the N_2_ injection. Hence, a thermochemical approach can reduce the required acid volume by around 23% compared to N_2_ injection.Sensitivity analysis indicates that the formation porosity and permeability have great impact on the acidizing treatment, while the pressure difference between the different layers has a minor effect on diversion performance.

## Figures and Tables

**Figure 1 molecules-25-02976-f001:**
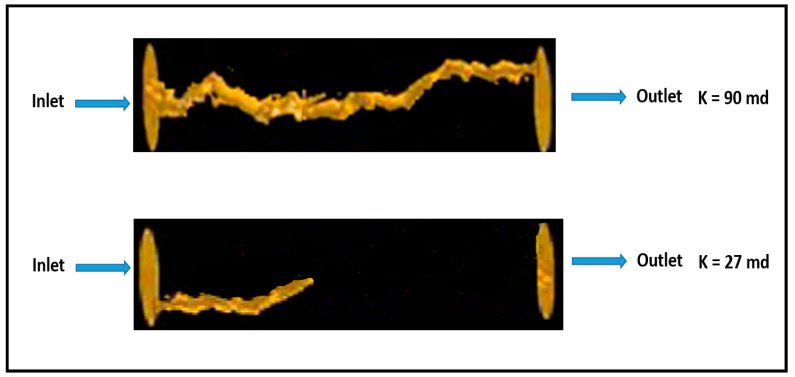
CT-scan images for the rock samples after acidizing treatment without using thermochemical fluids (no acid diverter).

**Figure 2 molecules-25-02976-f002:**
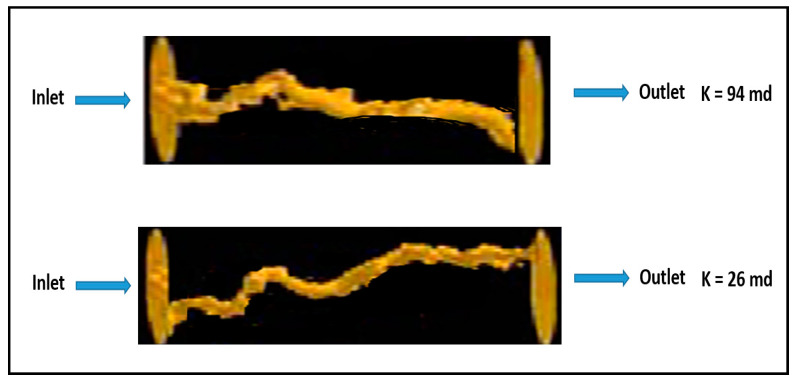
CT-scan images for the rock samples after diverting the injected acid using a thermochemical approach.

**Figure 3 molecules-25-02976-f003:**
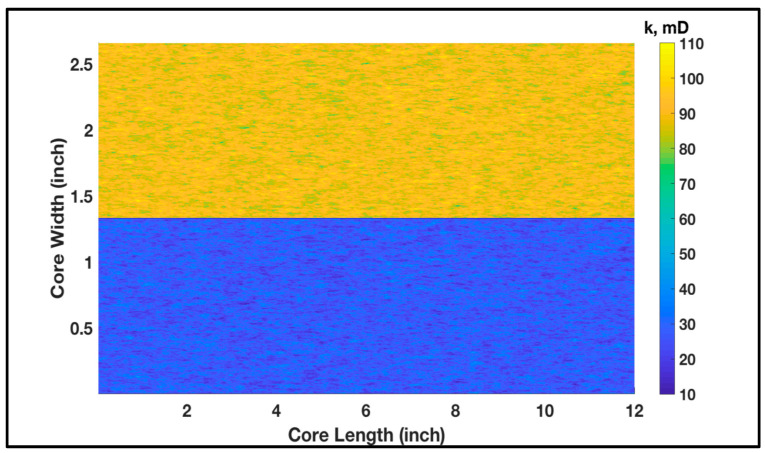
Spatially correlated permeability ($I_{x}=3.0,\;I_{y}=0.8,\;\sigma_{k}=3.0$) along the core dimensions before treatment.

**Figure 4 molecules-25-02976-f004:**
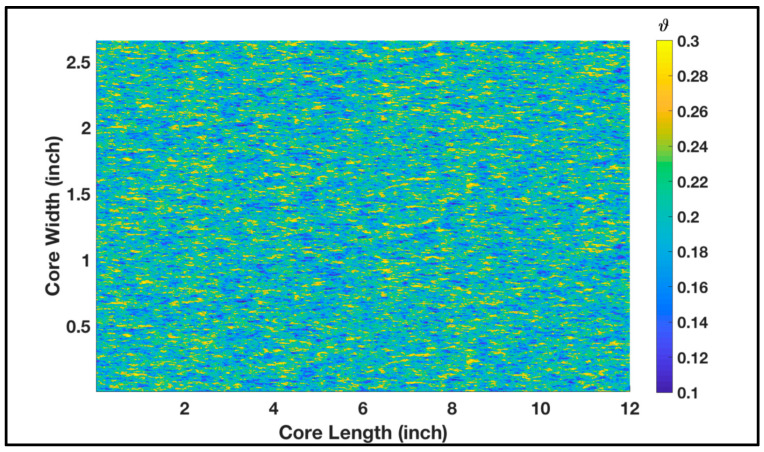
Spatially correlated porosity ($I_{x}=3.0,\;I_{y}=0.8,\sigma_{\varphi }=0.04$) along the core dimensions before treatment.

**Figure 5 molecules-25-02976-f005:**
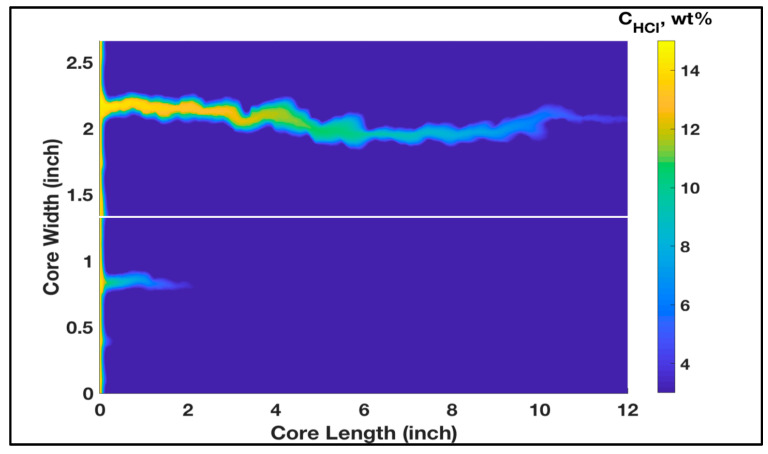
Concentration profile at the breakthrough for the HCl acid treatment with no diversion, showing a wormhole growing mainly in the high-permeability core sample.

**Figure 6 molecules-25-02976-f006:**
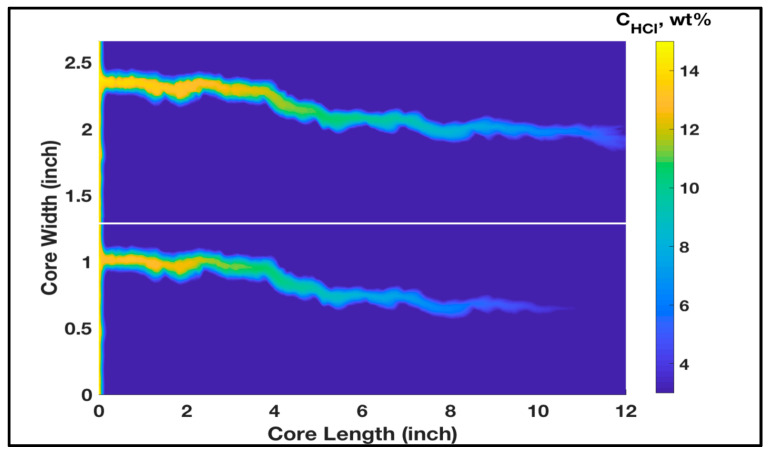
Concentration profile at the breakthrough for the thermochemical treatment case, showing a wormhole growing almost equally.

**Figure 7 molecules-25-02976-f007:**
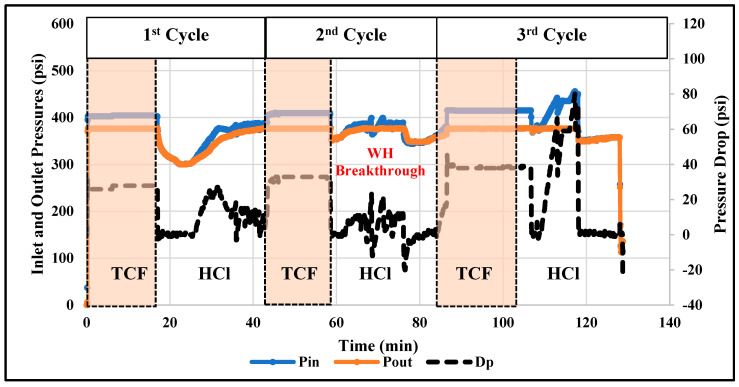
Profiles of inlet (Pin) and outlet (Pout) pressures and the pressure-drop across the low-permeability rock during alternating injection of thermochemical and HCl fluids. The pressure drop is read from the y-axis in the right-hand side.

**Figure 8 molecules-25-02976-f008:**
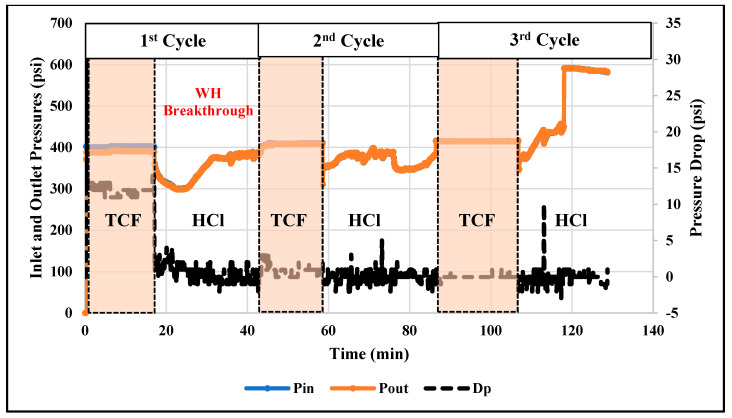
Profiles of inlet (Pin) and outlet (Pout) pressures and the pressure-drop across the high-permeability sample during the thermochemical/HCl treatment.

**Figure 9 molecules-25-02976-f009:**
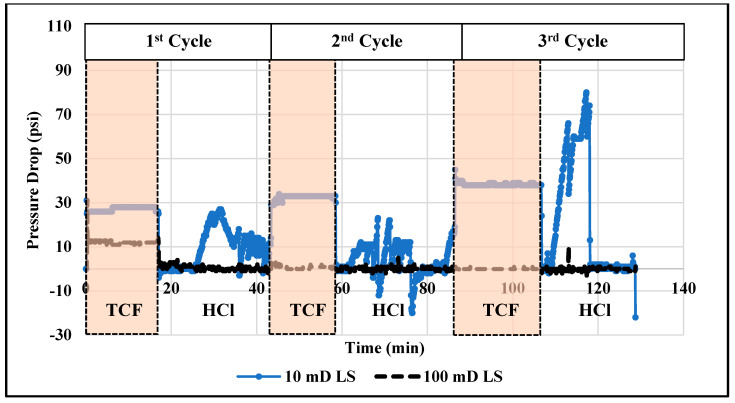
Profiles of pressure-drop across the low-permeability (10 mD) and high-permeability (100 mD) rocks during the alternating injection of thermochemical and HCl fluids.

**Figure 10 molecules-25-02976-f010:**
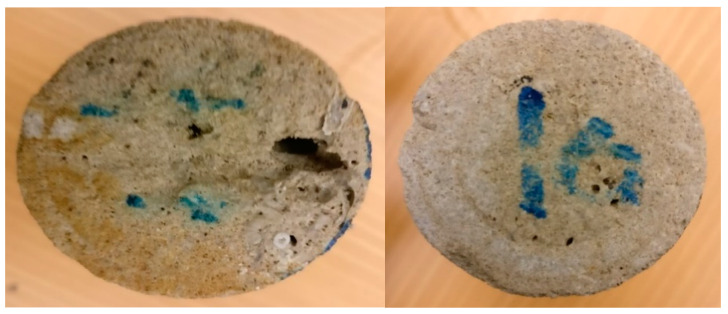
Images for the core inlet (**left**) and outlet (**right**), for the low-permeability sample after alternating treatment.

**Figure 11 molecules-25-02976-f011:**
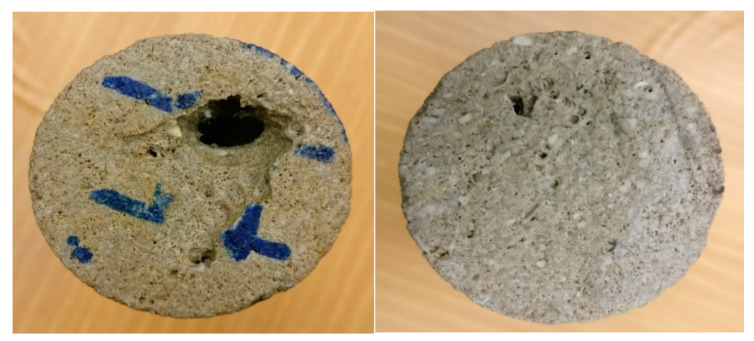
Images for the core inlet (**left**) and outlet (**right**) for the high- permeability sample after thermochemical/HCl treatment.

**Figure 12 molecules-25-02976-f012:**
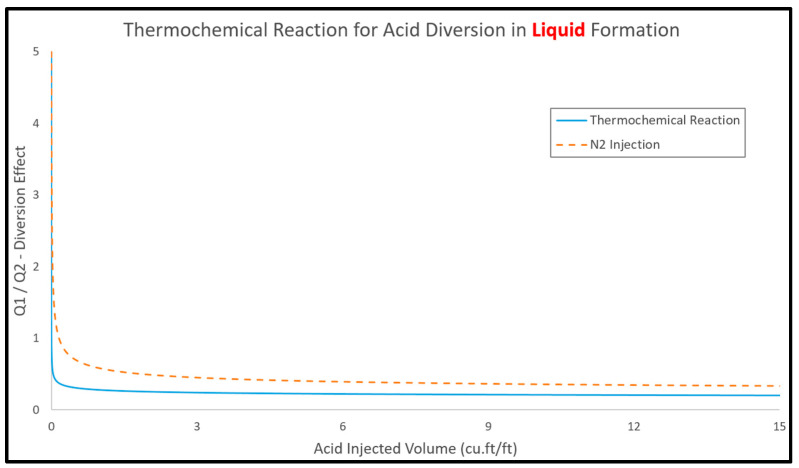
The injection rate ratio between top (Q1) and bottom (Q2) layer using thermochemical reaction and N_2_ injection.

**Figure 13 molecules-25-02976-f013:**
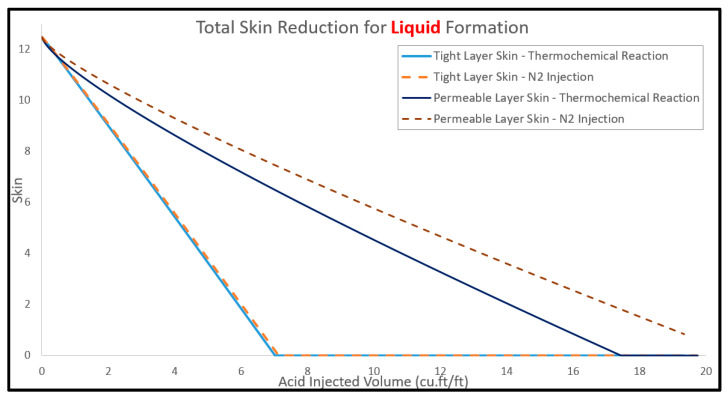
The skin behavior during acidizing and diversion process using thermochemical reaction and N_2_ injection.

**Figure 14 molecules-25-02976-f014:**
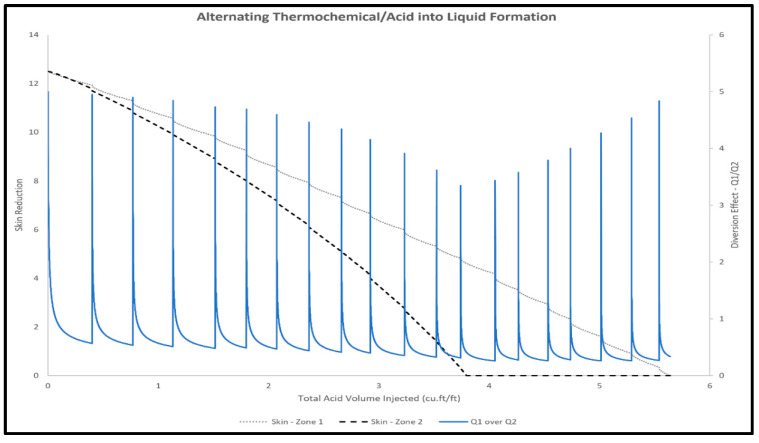
Skin reduction during alternating thermochemical and acid injection into high- (zone 1) and low-permeability layers (zone 2).

**Figure 15 molecules-25-02976-f015:**
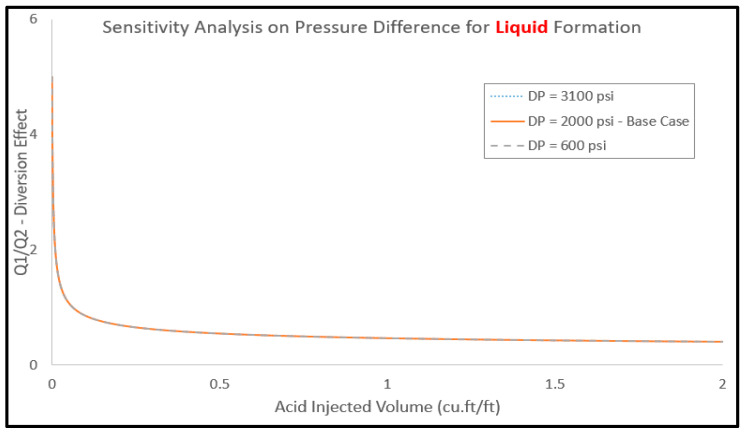
Effect of pressure difference (between layers 1 and 2) on the acid diversion performance.

**Figure 16 molecules-25-02976-f016:**
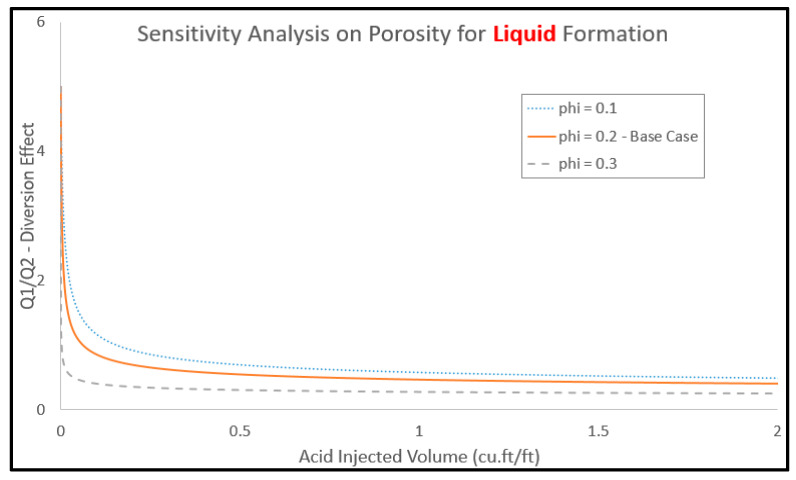
Effect of formation porosity on acid diversion performance.

**Figure 17 molecules-25-02976-f017:**
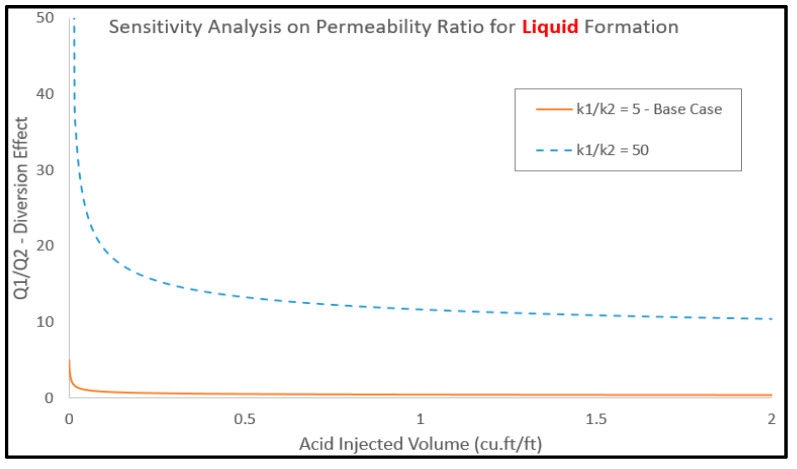
Effect of layer permeability on the acid diversion performance.

**Figure 18 molecules-25-02976-f018:**
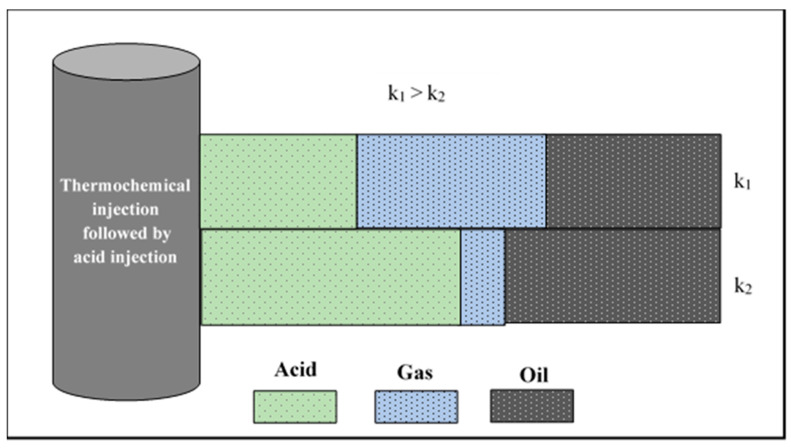
Acid diversion using thermochemical approach.

**Figure 19 molecules-25-02976-f019:**
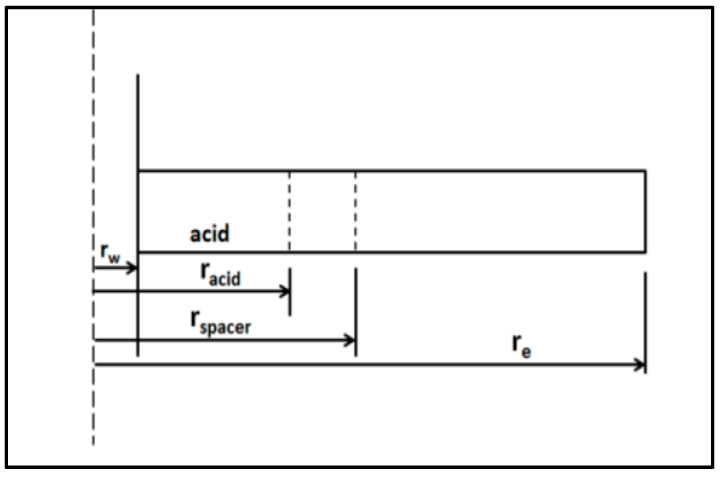
A schematic of the different regions around the wellbore during acidizing treatment.

**Figure 20 molecules-25-02976-f020:**
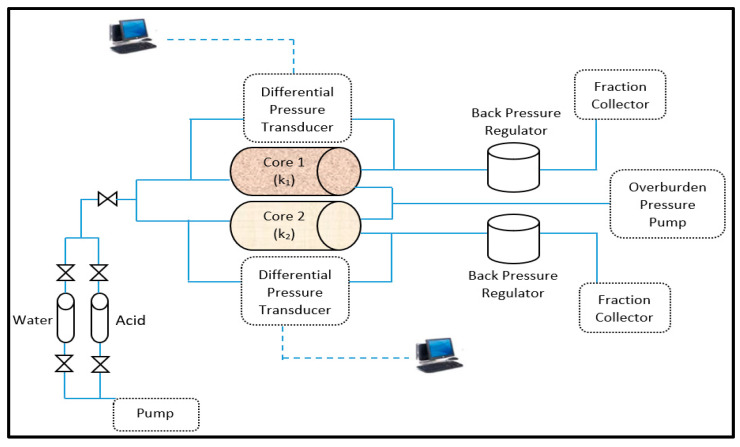
A schematic of the parallel coreflood setup.

**Table 1 molecules-25-02976-t001:** Reservoir, well, and fluid properties for liquid formation.

Parameter	Value
Porosity, ϕ (%)	20
Viscosity of reservoir fluid, μ_f_ (cP)	1
Viscosity of injected acid, μ_a_ (cP)	1.5
Viscosity of thermochemical fluids, (cP)	1.2
Wellbore pressure, p_w_ (psi)	3600
Pressure around wellbore, p_e_ (psi)	1600
Drainage radius, r_e_ (ft)	1676
Wellbore radius, r_w_ (ft)	0.25

**Table 2 molecules-25-02976-t002:** Input data for liquid formation.

Parameter	Layer 1	Layer 2
Permeability, k (mD)	100	20
Thickness, h (ft)	20	20
Damaged radius, r_d_ (in)	12	12
Skin, S_0_	12.5	12.5

**Table 3 molecules-25-02976-t003:** Model parameters for the sensitivity analysis scenarios.

Parameter	Base Case	Case 1	Case 2	Case 3	Case 4	Case 5
Pressure Difference, DP (psi)	2000	600	3100	2000	2000	2000
Porosity, ϕ (fraction)	0.2	0.2	0.2	0.3	0.1	0.2
Permeability ratio, k_1_/k_2_	5	5	5	5	5	50

**Table 4 molecules-25-02976-t004:** Properties of the rock samples used in this study.

Sample Index	Sample Type	Diameter (in)	Length (in)	Porosity (%)	Permeability (mD)
Core1	High-permeability rocks	1.5	12	20	94
Core2		1.5	12	20	90
Core3		1.5	4	19	102
Core4	Low-permeability rocks	1.5	12	20	26
Core5		1.5	12	20	27
Core6		1.5	4	18	11

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
