# Peer review of "A Novel Approach to Improve Acid Diversion in Carbonate Rocks Using Thermochemical Fluids: Experimental and Numerical Study"

_molecules, 2020, doi:10.3390/molecules25132976_

Round 1
Reviewer 1 Report
The manuscript presents a study of using thermochemical fluids injection to improve acid diversion near the wellbore in carbonate rock. Both laboratory experimental and numerical investigations have been conducted. The effectiveness of themochemical fluid injection for acid diversion improvement is justified through this study. This work has the practical importance to improve the conformance control for example in enhanced oil recovery for unconventional reservoir. I would love to see this work to be published. However, there are still somethings needed to be addressed before I can endorse a recommendation for a publication.
Below are some comments in details:
- The title is suggested to be "A Novel Approach to Improve Acid Diversion in Carbonate Rock Using Thermochemical Fluids: Experimental and Numerical Studies".
- In Line 23, it should be "continuous" instead of "continues"
- Remove "novel approach" from Keywords
- In Line 39, it should be "zones with large thickness"
- In Line 45, the "coiled tubing" should be acronymized when it appears at first time. In this case, it should be at Line 44. The acronym issue is here and there. For example, the "thermochemical" has been double acronymized at Line 177 and Line 181.This should be avoided.
- The literature review about using gas or foam for acid diversion is insufficient. How the proposed thermochemical fluid approach surpasses other gas approaches should be better addressed.
- In Line 83, replace "But" with "However"
- The novelty of thermochemical fluid injection should be better addressed.
- Where is it called "thermochemical" fluid injection? The "thermal part" of this method is not addressed. It has been briefly mentioned at Line 329-330. However, detailed explanation is needed. Otherwise, this method is more like "chemical fluid injection" approach.
- In Figure 4 and relevant numerical numerical simulations, why does the porosity distribution in the low permeability zone have no difference from high permeability zone? This should be explained or justified.
- In Line 172-193 and many other places (e.g., Line 206, 255, 256), the authors sometimes use "alternative" or "alternate". They all should be "alternating" injection.
- In Figure 7, "Pin", "Pout", and "Dp" should be defined firstly even though it is obvious. It should be pointed out in the caption that the Dp uses the right-hand side y axis. The text box "3rd Cycle" has covered some data which should be fixed.
- The schematic Figure 18 is suggested to be revised to make the concept clear. Current format of Figure 18 does not meet its purpose.
Author Response
Dear editor and reviewers, thanks for the efforts applied to review our manuscript. We appreciate the time spent and valuable comments. Please find below our response to your comments marked in red.
Reviewer 1
The manuscript presents a study of using thermochemical fluids injection to improve acid diversion near the wellbore in carbonate rock. Both laboratory experimental and numerical investigations have been conducted. The effectiveness of themochemical fluid injection for acid diversion improvement is justified through this study. This work has the practical importance to improve the conformance control for example in enhanced oil recovery for unconventional reservoir. I would love to see this work to be published. However, there are still somethings needed to be addressed before I can endorse a recommendation for a publication.
Below are some comments in details:
- The title is suggested to be "A Novel Approach to Improve Acid Diversion in Carbonate Rock Using Thermochemical Fluids: Experimental and Numerical Studies".
Thanks for the suggestion. The comment was addressed.
- In Line 23, it should be "continuous" instead of "continues"
Thanks for the note. The comment was addressed.
- Remove "novel approach" from Keywords
We agree. The key word was removed
- In Line 39, it should be "zones with large thickness"
Thanks for the correction. The comment was addressed.
- In Line 45, the "coiled tubing" should be acronymized when it appears at first time. In this case, it should be at Line 44. The acronym issue is here and there. For example, the "thermochemical" has been double acronymized at Line 177 and Line 181.This should be avoided.
Thanks for the observation. The manuscript was corrected accordingly.
- The literature review about using gas or foam for acid diversion is insufficient. How the proposed thermochemical fluid approach surpasses other gas approaches should be better addressed.
Thanks a lot, the literature review was improved, and the advantages of using thermochemical fluids for acid diversion were added in lines 72-78 and 93-98.
- In Line 83, replace "But" with "However"
The comment was addressed.
- The novelty of thermochemical fluid injection should be better addressed.
Thanks a lot, more texts were added in lines 99-103 to address this comment.
- Where is it called "thermochemical" fluid injection? The "thermal part" of this method is not addressed. It has been briefly mentioned at Line 329-330. However, detailed explanation is needed. Otherwise, this method is more like "chemical fluid injection" approach.
Thanks for the excellent comment. During thermochemical treatment, the primary mechanism for acid diversion is the in-situ gas generation which will block the high-permeability zones and divert the injected acids into the low-permeability zones, while the generated heat will have minor impact. However, for heavy oil reservoirs, the generated heat will significantly reduce the viscosity of reservoir fluids and then improve the hydrocarbon mobility. Overall, this method can be considered as chemical injection approach.
- In Figure 4 and relevant numerical simulations, why does the porosity distribution in the low permeability zone have no difference from high permeability zone? This should be explained or justified.
Thanks for the good observation. The lab measurements showed similar porosity. It is common to have similar porosity but different permeability. We added the following sentence to the text " Porosity lab measurements of the high and low permeability rock samples were similar; hence, the same distribution was assigned during simulations"
- In Line 172-193 and many other places (e.g., Line 206, 255, 256), the authors sometimes use "alternative" or "alternate". They all should be "alternating" injection.
Thanks for the observation. The manuscript was corrected accordingly
- In Figure 7, "Pin", "Pout", and "Dp" should be defined firstly even though it is obvious. It should be pointed out in the caption that the Dp uses the right-hand side y axis. The text box "3rd Cycle" has covered some data which should be fixed.
Thanks for the observation. The following was added to the caption "It is of note that the pressure drop is read from the y-axis in the right-hand side ". Also, the symbols "Pin", "Pout", and "DP" were defined, and the text box “ 3rd cycle” was fixed.
- The schematic Figure 18 is suggested to be revised to make the concept clear. Current format of Figure 18 does not meet its purpose.
Thanks a lot, Figure 18 was fixed.
Reviewer 2 Report
The article is a very well written work. The steps of the research are presented in detail.
Some small notes:
Please give an explanation of sigma_k in Figure 3 and sigma_fi in Figure 4.
In lines 422-423, the physical quantities should be indicated in italics.
The caption in Figure 19 should begin with a capital letter.
Based on MATLAB solutions, velocity and pressure distributions could be presented.
Author Response
Dear editor and reviewers, thanks for the efforts applied to review our manuscript. We appreciate the time spent and valuable comments. Please find below our response to your comments marked in red.
Reviewer 2
The article is a very well written work. The steps of the research are presented in detail.
Some small notes:
- Please give an explanation of sigma_k in Figure 3 and sigma_fi in Figure 4.
Thanks for the recommendation. The following was added " The standard deviation determines how far the distribution from the mean. The standard deviation for permeability distribution in the domain, , was 3 while for porosity, , was 0.04 as illustrated in the figures caption. "
- In lines 422-423, the physical quantities should be indicated in italics.
Thanks for the observation. The comment was addressed
- The caption in Figure 19 should begin with a capital letter.
Thanks for the observation. The comment was addressed
- Based on MATLAB solutions, velocity and pressure distributions could be presented.
Thanks a lot, the velocity and pressure distributions have similar trend to the skin factor, as indicated by equations (21,23, and 29). Also, in order to keep the paper within an acceptable length we prefer to not include them.
Reviewer 3 Report
The manuscript entitled “A Novel Approach to Improve Acid Diversion in Porous Media Using Termochemical Fluids: Simulation and Experimental Study” is an innovative contribution in the field of petroleum engineering. The authors propose to use a chemical reaction to produce in situ N2 gas to improve the acidizing treatment of carbonate reservoirs.
The experimental results obtained in the lab are quite promising. They demonstrate that alternate injections of the acidic and the reactive solutions greatly improve the acidizing treatment of low permeability rocks and lead a decrease of needed acid solution volume.
The method is interesting also under the environmental aspect because the product of the employed reaction is N2.
The experimental work is accompanied by MATLAB numerical simulations carried out in the framework of the two scale model, a reference model for this kind of investigations.
However, I found a few issues that require amendments.
1. The text is sometimes hard to read and I suggest a language check.
2. The parameters used in equations (4) and (5) should described more precisely, namely
CA starting acid concentration. Noteworthily the units used in the work, according to what is reported in the Glossary, differ from the units currently used in chemistry, i.e. mol/Lt
De: effective acid diffusion tensor? (missing in the Glossary)
β100 acid gravimetric dissolving power (capital B in the Glossary)
kr: the definition in the Glossary is confused: no HF is used in the present worn and the dimensions of a first order kinetic constant are concentration/time
3. I suggest to add the reference Panga, M.K.R., Ziauddin, M., Balakotaiah, V., 2005. Two-scale continuum model for simulation of wormholes in carbonate acidization. AIChE J. 51 (12), 3231–3248. https://doi.org/10.1002/aic.10574, which is a fundamental paper for the two scale model, here employed
4. I found opportune to add a comment about safety procedures for the handling of reactive solution that give rise to strongly exothermic reactions with sizable production of gas, albeit inert
5. Figures are rather large. Several of them can be reduced to half-size.
6. A general consideration. Throughout the paper not SI units were employed, rather US units?, common in petroleum engineering.
Author Response
Dear editor and reviewers, thanks for the efforts applied to review our manuscript. We appreciate the time spent and valuable comments. Please find below our response to your comments marked in red.
Reviewer 3
The manuscript entitled “A Novel Approach to Improve Acid Diversion in Porous Media Using Thermochemical Fluids: Simulation and Experimental Study” is an innovative contribution in the field of petroleum engineering. The authors propose to use a chemical reaction to produce in situ N2 gas to improve the acidizing treatment of carbonate reservoirs.
The experimental results obtained in the lab are quite promising. They demonstrate that alternate injections of the acidic and the reactive solutions greatly improve the acidizing treatment of low permeability rocks and lead a decrease of needed acid solution volume.
The method is interesting also under the environmental aspect because the product of the employed reaction is N2.
The experimental work is accompanied by MATLAB numerical simulations carried out in the framework of the two-scale model, a reference model for this kind of investigations.
However, I found a few issues that require amendments.
- The text is sometimes hard to read and I suggest a language check.
Thanks for the suggestion. We revised the text language.
- The parameters used in equations (4) and (5) should described more precisely, namely
CA starting acid concentration. Noteworthily the units used in the work, according to what is reported in the Glossary, differ from the units currently used in chemistry, i.e. mol/Lt
De: effective acid diffusion tensor? (missing in the Glossary)
β100 acid gravimetric dissolving power (capital B in the Glossary)
kr: the definition in the Glossary is confused: no HF is used in the present worn and the dimensions of a first order kinetic constant are concentration/time
Thanks for the observation. It was clarified that wt% was used in this study for acid concentration. The effective acid diffusion was added to the Glossary. The β100 was also fixed in the Glossary. For kr, it was changed to HCl instead of HF. We made the unit general to any exponent and that is why the term was raised to the power n.
- I suggest to add the reference Panga, M.K.R., Ziauddin, M., Balakotaiah, V., 2005. Two-scale continuum model for simulation of wormholes in carbonate acidization. AIChE J. 51 (12), 3231–3248. https://doi.org/10.1002/aic.10574, which is a fundamental paper for the two scale model, here employed
Thanks for the suggestion. The reference was added
- I found opportune to add a comment about safety procedures for the handling of reactive solution that give rise to strongly exothermic reactions with sizable production of gas, albeit inert
Thanks a lot, more texts were added in Lines 347-351.
- Figures are rather large. Several of them can be reduced to half-size.
Thanks for the suggestion. The figures were revised and reduced to the suitable sizes.
- A general consideration. Throughout the paper not SI units were employed, rather US units?, common in petroleum engineering.
True. It is common in petroleum engineering to use what we call field (US) units instead of SI. It will make it easier for a petroleum engineer to follow the context of this study.